# From Organotypic Mouse Brain Slices to Human Alzheimer’s Plasma Biomarkers: A Focus on Nerve Fiber Outgrowth

**DOI:** 10.3390/biom14101326

**Published:** 2024-10-18

**Authors:** Sakir Necat Yilmaz, Katharina Steiner, Josef Marksteiner, Klaus Faserl, Mathias Villunger, Bettina Sarg, Christian Humpel

**Affiliations:** 1Laboratory of Psychiatry and Experimental Alzheimer’s Research, Medical University of Innsbruck, 6020 Innsbruck, Austria; nyilmaz@mersin.edu.tr (S.N.Y.); k.steiner@i-med.ac.at (K.S.); 2Department of Histology and Embryology, Faculty of Medicine, Mersin University, Mersin 33130, Turkey; 3Department of Psychiatry and Psychotherapy A, Hall State Hospital, 6060 Hall in Tirol, Austria; josef.marksteiner@tirol-kliniken.at; 4Protein Core Facility, Institute of Medical Biochemistry, CCB-Biocenter, Medical University of Innsbruck, 6020 Innsbruck, Austria; klaus.faserl@i-med.ac.at (K.F.); mathias.villunger@i-med.ac.at (M.V.); bettina.sarg@i-med.ac.at (B.S.)

**Keywords:** Alzheimer’s disease, biomarker, plasma, organotypic brain slice, nerve fiber, myelin oligodendrocyte glycoprotein, microcontact printing, mass spectrometry, immunofluorescence labeling

## Abstract

Alzheimer’s disease (AD) is a neurodegenerative disease characterized by memory loss and progressive deterioration of cognitive functions. Being able to identify reliable biomarkers in easily available body fluids such as blood plasma is vital for the disease. To achieve this, we used a technique that applied human plasma to organotypic brain slice culture via microcontact printing. After a 2-week culture period, we performed immunolabeling for neurofilament and myelin oligodendrocyte glycoprotein (MOG) to visualize newly formed nerve fibers and oligodendrocytes. There was no significant change in the number of new nerve fibers in the AD plasma group compared to the healthy control group, while the length of the produced fibers significantly decreased. A significant increase in the number of MOG+ dots around these new fibers was detected in the patient group. According to our hypothesis, there are factors in the plasma of AD patients that affect the growth of new nerve fibers, which also affect the oligodendrocytes. Based on these findings, we selected the most promising plasma samples and conducted mass spectrometry using a differential approach and we identified three putative biomarkers: aldehyde-dehydrogenase 1A1, alpha-synuclein and protein S100-A4. Our method represents a novel and innovative approach for translating research findings from mouse models to human applications.

## 1. Introduction

### 1.1. Identifying CSF and Plasma Biomarkers in Alzheimer’s Disease

Alzheimer’s disease (AD) is a devastating pathology that occurs in older ages and is characterized by the progressive loss of cognitive functions and memory. In parallel with the developments in modern medicine, the rate of disease occurrence in populations is rising as human life expectancy increases. Over 55 million people worldwide are living with dementia, 60–70% of whom have AD, and their number is increasing [1,2]. In addition to the lack of an effective treatment for AD, one of the greatest challenges is that it is not possible to diagnose the disease early. The disease can be diagnosed almost entirely with clinical examination and cognitive tests. However, when the diagnosis is made, it is usually in an advanced stage. A definitive diagnosis can only be made by demonstrating post-mortem beta-amyloid (Aβ) plaques and tau neurofibrillary tangles in brain tissue. Unfortunately, there are no reliable laboratory tests that allow early diagnosis before the onset of clinical findings of the disease. Currently, four biomarkers, namely Aβ-42, Aβ-40, total tau and phosphorylated (phospho-)tau-181, are widely accepted in cerebrospinal fluid (CSF) [3]. According to the National Institute on Aging and Alzheimer’s Association (NIA-AA) guidelines, the use of biomarkers in the diagnosis of AD is optional and helps to support the diagnosis rather than to make a diagnosis. No consensus has yet been reached on studies conducted in easier to obtain and low-risk fluids such as plasma. Although plasma levels of Aβ or phospho-tau can be measured with high accuracy and sensitivity, the correlation of the results with AD is considered weak [4,5].

In addition to the biomarkers mentioned above, biomarker candidates, either individually or in panels, have been investigated for a long time [3,6,7,8,9]. Many different studies have suggested different biomarkers, and one study even claimed 18 different molecules as biomarkers, but none of them have been validated and accepted to date [10]. Along with measuring the levels of specific molecules, changes in the ratios of these levels are also suggested as biomarkers [11]. However, almost none of these and similar studies have reached results that are accepted today. They also require difficult and expensive methods such as mass spectrometry. Today, the most widely accepted parameter with this method is the lower plasma Aβ42/Aβ40 ratio [12,13] and plasma phospho-tau levels [14]. The identification and availability of specific and reliable biomarkers in body fluids for early diagnosis of AD is still one of the most important priorities of this disease and has the potential to radically change the prognosis of AD [15,16].

### 1.2. Utilizing Organotypic Brain Slices to Identify Biomarkers

The vast majority of studies on biological processes, diseases and treatments are performed using in vitro cell culture or in vivo animal experiments. However, primary cell cultures cannot provide the three-dimensional architecture of the organ and the interaction characteristics of different types of cells, while animal experiments have disadvantages such as the necessity of using a large number of animals and procedural challenges. Organotypic brain slices stand out as a method that largely eliminates the disadvantages of these two methods and combines their advantages [17]. With this groundbreaking technique, 150-µm-thick slices can be taken from postnatal-day-8-old mice without losing their viability and very different and valuable experimental designs can be applied by keeping them alive for weeks to months. In this method, the molecules to be studied are usually added directly to the culture medium or adsorbed onto some biomaterials and applied to the slices. As an alternative, we developed a microcontact printing (µCP) method, where molecules can be printed as intermittent lines on membranes at micrometric resolution using collagen as scaffold and the slices are cultured along these prints. In this way, the effect of the molecule on the cells in the slices can be observed directly based on evidence. Using this method, we have successfully studied the activation of microglia [18], blood vessel development [19], or the effects of nerve growth factor (NGF) on cholinergic neurons [20].

### 1.3. Nerve Fibers and Neurodegeneration in Alzheimer’s Disease

The major two pathologies in AD are the accumulation of misfolded Aβ plaques and hyperphosphorylated tau protein tangles within the brain and there are different hypothesis about their formation including the redox-dependent copper effect [21] or the calcium hypothesis [21,22]. While these hallmark features are well-documented, recent research has increasingly focused on the role of nerve fibers and neurodegeneration in the pathogenesis of AD [23,24,25]. Nerve fibers, particularly axons and dendrites forming the intricate neuronal networks are essential for transmitting signals throughout the brain. In AD, these nerve fibers undergo significant structural and functional alterations that contribute to the cognitive impairments observed in patients. Studies using advanced neuroimaging techniques such as magnetic resonance imaging (MRI) have revealed abnormalities in white matter integrity, indicating early and widespread axonal damage in AD brains [26]. Recent research suggests that nerve fiber alterations are not only a consequence of a diseased environment but may actively contribute to the spread of neurodegeneration in AD. As evidence for this hypothesis, it has been shown in various studies that axonal transport is impaired at an early stage due to the accumulation of Aβ [27]. Cytotoxic mediators released from damaged axons into the environment damage other neurons and axons, and axonal damage findings are detected before neuronal death occurs [28,29]. Considering these results, early numerical and structural abnormalities of nerve fibers may be a valuable parameter in the early diagnosis of AD.

In the present study, we continue our work on human plasma-derived factors on microglia in brain slices [30] and we hypothesize that plasma from AD patients may also contain or lack factors compared to healthy subjects, which could differentially influence the neurofilament (NF)+ nerve fiber growth in mouse brain slices. To test our hypothesis, we labeled brain slices with NF antibody to show the development and numbers of nerve fibers. We also used the myelin oligodendrocyte glycoprotein (MOG) antibody, an oligodendrocyte marker, because of the well-known close relationship of the myelin sheath with axon development, function and survival [31]. Thus, we aimed to investigate the possible effects of increased or decreased factors in the plasmas of study groups on nerve fiber development related to oligodendrocytes. We performed mass spectrometry on the most characteristic samples to identify new potential biomarkers.

## 2. Materials and Methods

### 2.1. Human Plasma Samples

Human blood plasma samples were collected from 3 different groups: healthy controls (*n* = 17; 6 male, 11 female) and mild cognitive impairment (MCI) (*n* = 18; 6 male, 12 female) and AD patients (*n* = 17; 6 male, 11 female). Subjects were selected from Caucasian individuals aged over 60. Approval for the study was received from the local ethics committee of Innsbruck Medical University (AN2015.0159 351/4.7 405/5.5 (4484a), and the principles of the Declaration of Helsinki were followed. All subjects signed a written informed consent form. The subjects were selected from among patients treated at the Landeskrankenhaus Hall/Tirol, Austria, under the supervision of Prim. Univ.—Prof. Dr. Josef Marksteiner. Plasma samples collected since 2014 were stored at −80 °C without thawing. Samples with similar storage times were used in the study to ensure that the degradation effect due to storage time did not disrupt the standard homogeneity.

Patients were diagnosed as described in detail in a previous study [30]. Geriatric Depression Scale (GDS) and the Mini Mental State Examination (MMSE) tests were applied to the patients, and plasmas of patients with a definitive diagnosis of AD and MCI were used in this study. Magnetic resonance imaging of the patients was performed with a 1.5 Tesla Siemens Symphony MRI device. Patients with severe medical or neurological deficits that might confound the study groups were excluded from this study. Participants underwent continuous statin or ezetimibe treatment for at least 3 months before study entry. No patient had a cholesterol level > 240 g/dL that was not treated with a statin or ezetimibe. Lifestyle as a factor is rather challenging to measure. After the patients were identified, 10 mL of blood was taken from each patient into tubes containing EDTA and the blood was processed within 24 h. Samples were centrifuged at 2300× *g* for 5 min and the upper plasma phase was taken and stored at −80 °C (Figure 1).

### 2.2. Microcontact Printing of Human Plasma

Detailed information about the collagen hydrogel-based microcontact printing method can be found in our previously written comprehensive methodological review [32]. A master mold of 50 µm wide, 800 µm long, 50 µm apart and 50 lines was custom prepared and purchased from GeSiM “www.gesim.de (accessed on 18 August 2024)” (Figure 1). By using this mold, polydimethylsiloxane stamps (PDMS, Sylgard 184 Silicone Elastomer Kit, Dow, Seneffe, Belgium, 01673921) fabricated according to the method mentioned above [32]. The microcontact printing method was briefly carried out as follows: to prepare plasma printing solution, 67 µL of bovine collagen solution type I (stock solution of 3 mg/mL, Sigma-Aldrich, St. Louis, MO, USA, 804592), 10 µL of 100 mM phosphate-buffered saline (PBS, pH 7.4, sterilized), 5 µL of 10 mM PBS and for assessing the printing efficiency, 5 µL of red-fluorescent Alexa Flour 546 anti-rat antibody (Invitrogen, Thermo Fisher Scientific, Waltham, MA, USA, A11081) were added to the lyophilized plasma (100 µL) respectively, and mixed by vortexing until completely dissolved. 1N 0.8 µL NaOH solution was added to final solution for adjusting the pH to 7.2. As a final step, 12.5 µL of the crosslinker 4arm-poly(ethylene glycol) (PEG) succinimidyl succinate (stock solution of 12.5 mg/mL, Sigma-Aldrich, St. Louis, MO, USA, JKA7006) was added to the prepared solution just before starting the printing process. Then, 15 µL of this prepared mixture was dropped onto the patterned surface of each stamp and covered with a coverslip to ensure even distribution on the stamp surface. Stamps were incubated at 37 °C for 15 min and the coverslips were removed. Excess solution was removed from the stamp surfaces using the coverslip edge. After the stamp surface was completely dry, in order to transfer the solution to a semi-permeable membrane, the stamps were placed with their patterned surface facing the membrane and 18 g weights were placed on them and left at RT for 1 h. When the time was up, the weights were carefully removed and the stamps were left on the membranes O/N at +4 °C. The next morning, the stamps were separated from the membrane and the printing activity was evaluated under a fluorescent microscope by visualizing Alexa Fluor 546 anti-rat antibody. Membranes with insufficient printing were not used in this study (Figure 1). In order to place the brain slices in the correct position on the prints, a small reference point was placed on the same edge of the print areas of all membranes with a marker pen. After the membranes were sterilized under UV light for 20 min, they were kept in the incubator with sterile medium for at least 15 min for equilibration (Figure 1).

### 2.3. Organotypic Brain Slices

Preparation of organotypic half brain slices was described comprehensively in our previous study [32]. In brief, 8–10-day old C57BL/6 wild-type mice were decapitated rapidly and their brains removed aseptically. The brains were glued to the holder with their frontal parts facing up with a cyanoacrylate glue (Loctite 401, Henkel, Düsseldorf, Germany, 231435). By using a water-cooled vibratome (Leica, VT1000S, Nussloch, Germany) 150 µm thick coronal brain slices were taken at the hippocampal level in sterile medium. Slices were halved horizontally and the upper parts containing the hippocampus were placed on microcontact-printed semipermeable extra membranes with a pore size of 0.4 µm (Isopore, Merck Millipore, Darmstadt, Germany, HTTP02500) in cell culture inserts (Millicell, Merck Millipore, Darmstadt, Germany, PICM03050) (Figure 1). Organotypic brain slice culture was performed in 6-well plates (Sarstedt, Nümbrecht, Germany, 83.3920) filled with 1 mL of medium (pH 7.2, 1× MEM (Gibco, Thermo Fisher Scientific, Waltham, MA, USA, 11012044), 5.12 mM NaHCO_3_ (Merck Millipore, Darmstadt, Germany, 106329), 31.5 mM glucose (Merck Millipore, Darmstadt, Germany, 49159), 2 mM glutamine (Merck Millipore, Darmstadt, Germany, 100289), 10% heat-inactivated horse serum (HS, Gibco, Thermo Fisher Scientific, Waltham, MA, USA, 16050-122), 0.25× HBSS (Gibco, Thermo Fisher Scientific, Waltham, MA, USA, 24020091), 1× antibiotic-antimitotic solution (Sigma-Aldrich, St. Louis, MO, USA, A5955). Slice culture was carried out at 37 °C in a 5% CO_2_ environment for two weeks by changing medium once a week. Slices were fixed for 3 h in 4% PFA at 4 °C at the end of the culture period. After fixation, all slices were washed with PBS and stored in PBS containing 0.1% NaN_3_ for later use.

All animal experiments were approved by the Austrian Ministry of Science and Research and conducted in accordance with Austrian animal welfare and experimentation guidelines. All animal experiments, according to Austrian laws, were classified as “Organentnahme”.

### 2.4. Immunofluorescence for Nerve Fibers: NF, MAP2 and MOG

Immunofluorescence labeling was performed according to the previously described method [32]. Fixed slices were permeabilized with 0.1% T-PBS (Triton X100-PBS) for 30 min by gentle shaking at RT. Following washing with 10 mM PBS 3 × 3 min, slices blocked with T-PBS 0.1% + BSA 0.2% + HS 20% by shaking at RT for 30 min to avoid unwanted background staining. Then, blocking solution was withdrawn without washing, and slices were incubated with primary antibody mixture against neurofilament H (NF-H, Synaptic Systems, Göttingen, Germany, Cat No: 171106, 1:1000), microtubule-associated protein-2 (MAP2; Synaptic Systems, Göttingen, Germany, Cat No: 188006, 1:1000) or myelin oligodendrocyte glycoprotein (MOG, Proteintech, Rosemont, IL, USA, Cat No: 12690-1-AP, 1:1000) diluted in T-PBS 0.1% + BSA 0.2% for 48 h at 4 °C. Slices were washed 3 × 3 min with PBS and incubated Alexa Fluor 488 conjugated anti-chicken (MAP2 or NF-H) or Alexa Fluor 546 conjugated anti-rabbit (MOG) secondary antibody (Invitrogen, Thermo Fisher Scientific, Waltham, MA, USA, Cat No: A-21206 and Cat No: A-10040, 1:400) and counterstained with nuclear blue-fluorescent dye DAPI (Sigma-Aldrich, St. Louis, MO, USA, D9542, 1:10,000) diluted in T-PBS 0.1% + BSA 0.2% at RT for 1 h with shaking. During and after incubation slices were protected from light. Slices were washed 3 × 3 min with PBS and mounted on glass slides with mowiol. Imaging and photographing were performed with a fluorescence microscope (Olympus, Tokyo, Japan, BX61) using green channel for Alexa 488 (ex 480/40 nm, em 527/30 nm) and red channel for Alexa 546 (ex 535/50 nm, em 610/75 nm) by using a camera control and imaging software connected to the microscope (Openlab software, version 5.5.0, Improvision, Coventry, England).

### 2.5. Mass Spectrometry 

To prepare the samples for analysis, 14 highly abundant proteins were removed from the plasma by depletion. For this purpose, High Select™ Depletion Spin Columns (P/N: A36369, Thermo Scientific, Rockford, IL, USA) were used according to the manufacturer’s instructions. The proteins in the flow-through in the 400 µL sample volume were reduced with 40 µL of dithio-threitol prepared as 100 mM in PBS buffer for 30 min. Then, for the alkylation of free cysteines, 40 µL of 550 mM iodacetamide prepared in PBS buffer was applied for 20 min at room temperature in the dark. Samples lyophilized to 200 µL were diluted 5-fold with acetonitrile, vortexed and centrifuged at 16,000× *g* for 5 min. The pellet, washed twice with ethanol, was dissolved in 100 µL of 100 mM TEAB buffer (pH 8.5). Proteins were digested O/N at 37 °C with agitation using 1 µg of trypsin (Sequencing Grade Modified Trypsin, P/N: V5111, Promega). Digested peptides were TMT-labeled according to the manufacturer’s instructions (TMT10plex™ Label Reagent Set, P/N A58332, Thermo Scientific, Waltham, MA, USA). The pooled samples were dried by lyophilization and redissolved in 85 µL of 0.1% formic acid. Peptides were then fractionated by high pH reversed phase chromatography using the XBridge Peptide BEH C18 column, 4.6 mm × 250 mm, 300 Å, 5 µm (P/N 186003625, Waters, Milford, MA, USA) as previously described [33].

Liquid chromatography system coupled with tandem mass spectrometry (nanoLC-MS): Digested peptides were analyzed using a liquid chromatography system (UltiMate 3000 nano-HPLC, Thermo Scientific, Waltham, MA, USA) coupled to a mass spectrometer (Orbitrap Eclipse, Thermo Scientific, Waltham, MA, USA) as reported in a previous study [34]. Briefly, peptides were eluted using an acetonitrile gradient with a total gradient time of 142 min to remove salts and buffers on a homemade column (100 μm i.d. × 17 cm long) packed with 2.4 μm C18 material (Reprosil, Dr. A. Maisch HPLC GmbH, Ammerbuch-Entringen, Germany). The Orbitrap Eclipse mass spectrometer was operated in data-dependent mode with a three-second cycle time. Survey full scan MS spectra were obtained at a resolution of 120,000, while MS2 spectra were obtained at a resolution of 50,000. Fragmentation was accomplished by higher energy collisional dissociation with a normalized collision energy of 38.

MS Database search: The MS data were analyzed using Proteome Discoverer software (V 3.1 Thermo Scientific, Waltham, MA, USA). MS/MS spectra were searched against the Uniprot human reference proteome database (last updated 27/03/2024) by using Sequest HT engine. The search was performed with the following parameters: Enzyme specificity was set to allow up to two miscleaveages for trypsin. Carbamidomethyl on cysteine was the fixed modification, while oxidation of methionine was the variable modification. The precursor mass tolerance was set as 10 ppm, while the fragment mass tolerance was 20 mmu. Maximum false discovery rate (FDR) was set to 1% for protein and peptide identification. The protein fold changes were calculated for quantitation, according to TMTpro reporter ion intensities found in MS2 scans. Sequence similarity between human and mouse were searched via the NCBI BLAST+ software (Version: 2.16.0) and database (EMBL’s European Bioinformatics Institute, Hinxton, Cambridgeshire) [35].

### 2.6. Data Analysis and Statistics

Sample size was calculated according to Cohen [36] calculating a type I error of 5% and a test power of 95% and a minimal sample number of 50 cases. All quantitative analyses were performed blindly. Only nerve fibers growing outward from the slice and located on the microcontact print lanes were included in the evaluation. The number and length of the NF positive nerve fibers and MOG positive dot numbers around these fibers were evaluated. The number of nerve fibers and dots was assessed in an area 300 µm long or 6 microcontact-printed lanes wide along the horizontal cut edge. Measurements and counts were performed using manual cell counter tool of ImageJ software (version 1.54; National Institute of Health, Bethesda, MD, USA). The average number of NF positive nerve fibers per mm was calculated for each slice. To measure the nerve fiber lengths, software was calibrated by calculating pixel-to-µm ratio using a micrometric ruler. MOG positive dots were calculated as the average number of dots per nerve fiber. The number of microcontact-printed plasmas obtained from different patients or individual animals used was considered as the sample size (*n*). All data are presented as mean ± standard error of the mean (SEM). Statistical analysis was performed with one-way ANOVA followed by a Fisher LSD post hoc test, with *p* < 0.05 considered significant.

## 3. Results

### 3.1. Epidemiology of the Patients Used in This Study

In the present study we included 17 healthy controls, 18 MCI and 17 AD patients, and achieved a balanced distribution of gender, with 6 males and more females (Table 1). The mean age of the patients was approximately 73 years, and this mean was slightly higher in the AD group (Table 1). The patients’ MMSE scores were 29.6 in the control group, significantly lower in MCI (27.5), and markedly decreased in AD (18.3) (Table 1). The Geriatric Depression Scale Score was 2.6 in the controls, 3.7 in the MCI group and 2.5 in the AD groups, and no differences were seen (Table 1).

### 3.2. NF and MAP2-Positive Nerve Fibers in Cultured Organotypic Brain Slices

In order to demonstrate outgrowing nerve fibers in the cultured brain slices, anti-neurofilament (NF) antibodies were used. Anti-MAP2 antibodies were also used for confirmation of nerve fiber labeling specificity. We performed a double labeling protocol for NF and MAP2 antibodies and evaluated co-localization of these antibodies. Our results show that the NF antibody specifically labelled nerve fibers and co-localized with MAP2 (Figure 2).

### 3.3. MOG-Positive Nerve Fiber Dots on NF-Positive Nerve Fibers

When NF-positive fibers extending from the lower edge of the slices were examined, some MOG positive structures in the form of dots were observed in close contact with them. These dots were located along the nerve fibers as scattered structures without continuity. In a few samples, these dots were also observed to be located around the neuronal bodies (Figure 3).

### 3.4. Effects of Human Plasma on NF+ Mouse Nerve Fiber Growth

After two weeks of culture, new formed nerve fibers extending from the lower edge of the brain slices along the microcontact-printed lanes were counted and measured. Compared to the healthy control plasma-printed group, the plasma-free group showed a statistically significant increase in the number of new nerve fibers (Table 2). However, when the lengths of newly formed nerve fibers were evaluated, the mean fiber length of this group was significantly decreased compared to the healthy control (Table 2). On the other hand, there was no significant difference between the mean number of newly formed fibers in the MCI and AD groups and the mean number of fibers in the healthy control group (Table 2). While no significant difference was found between the MCI group and the healthy control group in terms of fiber length means, in the AD group fiber lengths were significantly decreased compared to the control group (Table 2).

### 3.5. Effects of Human Plasma on MOG+ Dots Per Nerve Fiber

Considering that the numbers of the MOG+ dots may differ between different groups, they were counted and calculated as the number of dots per fiber. The group means of the counting results were calculated and it was evaluated whether there was a statistically significant difference between the groups. The mean of all groups was compared with the mean results of plasma samples of healthy individuals (Table 2). It was observed that the number of MOG+ dots around the nerve fibers extending from the slices cultured on plasma-free microcontact prints was significantly lower than in the microcontact print samples containing plasma of the healthy control group (Table 2). On the other hand, there was an increase in MOG positive dots in proportion to the severity of the disease in the MCI and AD groups (Table 2). Although the difference in increase between the healthy group and the MCI group was not significant, there was a significant increase in the AD group compared to the control group (Table 2) (Figure 4).

### 3.6. Mass Spectrometry

We performed mass spectrometry from the top three control and top three AD samples selected according to the results in Table 2 (Figure 5). Following depletion of the 14 most prominent proteins in plasma (e.g., albumin, immuno-globulins, fibrinogen, transferrin, …), we quantitatively measured approximately 1400 proteins with an overlap of 1373 proteins (Figure 5).

With these results, we further restricted the number of proteins using the keywords “Oligodendrocyte” or “axon” or “myelin” and identified three possible biomarkers: aldehyde-dehydrogenase 1A1, alpha-synuclein and protein S100-A4 (Table 3).

## 4. Discussion

In the present study, our aim was to investigate the effects of possible plasma factors on nerve fibers using mouse organotypic brain slice culture combined with human plasma-loaded microcontact prints. After the culture period, we observed the proliferation and growth of nerve fibers as well as the change in MOG+ oligodendrocytes. Our findings show that plasma samples taken from AD patients did not cause significant changes in the number of new nerve fibers, but significantly reduced the lengthening of nerve fibers and enhanced MOG+ dots per fiber. Using the most promising samples, we identified three putative biomarkers: aldehyde-dehydrogenase 1A1, alpha-synuclein and protein S100-A4.

### 4.1. Organotypic Mouse Brain Slices and µCP of Human Plasma

Organotypic brain slice culture has been used for many years as a technique that combines the advantages of cell culture and animal experiments, allowing in vivo experiments to be performed under in vitro conditions. In a 150-µm-thick slice, all features of the brain, such as cytoarchitecture, proliferation, differentiation, intercellular interactions and functionality, are almost completely preserved [17,18,19,20,23,32,37,38,39]. Although the brain slices obtained have lost their blood circulation, studies have shown that the structural cells of the blood vessels maintain their organization and functions to a significant extent. In other words, blood vessels can largely preserve their histological structure, intercellular connections and paracrine functions in vitro [40,41].

Culturing of brain slices has been used effectively in our laboratory for a long time and more than 60 studies have been carried out and published. Studies in the literature and our own experience show that slices obtained from postnatal 8–10-day-old mice have the highest viability and resistance to mechanical traumatic damage in vitro [42]. Therefore, we used postnatal 8-day-old animals in the present study. Slice culture was performed using the culture technique at the liquid–air interface on a semipermeable membrane in a humidified environment at 37 °C as described [43]. During the two-week culture process, slices were observed to become flat and transparent to ensure their viability and quality [17].

One of the most important advantages of the organotypic brain slice culture method is that a large number of slices (8–10 for the hippocampus region in this study) can be obtained from each animal. This number can be much higher for more widespread tissues such as the entire cerebral cortex [39]. Thus, slices from the same animals can be used in different experimental methods to provide a high level of standardization for the samples and at the same time it is also compatible with the 3R principle of animal testing as it reduces the number of animals used.

In past studies, attempts have been made to transfer drugs, proteins, genes and cells to slice cultures by either adding them directly to the medium or by loading them onto various synthetic or natural biomaterials [44,45]. The microcontact printing technique is a method in which the desired molecules are immobilized onto a suitable material in the form of micrometric scaled patterns for providing them to living cells and tissues [46,47]. In our laboratory, a modified microcontact printing method that enables the transfer of proteins by printing them in a micrometric scaled pattern to living brain slices has recently been developed and optimized [32]. It is an attractive and promising method due to its precision at the micrometric scale and its easy preparation, and it is a reliable method to evaluate the effects of different molecules on cells. Our studies on microglia [18], brain vascular structures [19] and nerve fibers [20] showed that the method is highly effective and is a suitable model for many different experimental designs in this field. Microcontact printing, a soft lithography technique, is prepared as a soft elastomeric stamp. Due to this structure, it is highly biocompatible and suitable for transferring almost any type of protein to a substrate [48]. A suitable structural support is needed for cells and tissues to be organized in a functional manner. In living organisms, this function is provided by the extracellular matrix (ECM), a substance that fills the spaces between cells. The ECM is a network structure composed of proteins and polysaccharides secreted from local cells and functions as a natural scaffold for tissues and organs. It regulates functions such as proliferation, differentiation, migration and cell death by providing adhesion, communication and substance exchange between cells [49]. Many biomaterials such as Matrigel, fibrin, polypeptides, hyaluronic acid, various polymers and collagen that can be used instead of ECM have been developed and used successfully in in vitro 3D culture experiments. Among these, collagen is one of the most widely used materials [50,51,52]. Collagen can be manufactured either as a cross-linked compact solid or as a lattice-like hydrogel. These synthetically formed collagens, which have bioactive and biodegradable properties, provide appropriate structural and biochemical support for cells and tissues [53]. In a previous study, we described a well-designed delivery system based on collagen, chemically crosslinked with 4-arm PEG that releases loaded bioactive molecules in a controlled manner [54]. Although collagen is not typically found in the parenchymal tissue of the central nervous system and is found only in the vascular basement membranes, it has been shown to be highly effective on the physiological and pathological behavior of neurons [55,56]. Consistent with the literature, we observed in our findings that collagen alone has significant neurobiological effects. In vitro, microcontact printed collagen promoted the increase in the number of new nerve fibers.

### 4.2. New NF+ and MAP2+ Nerve Fibers Grown from Mouse Brain Slices

NFs, neurotubules, microfilaments and their associated proteins, are the cytoskeletal elements of neurons and play essential roles in the acquisition and maintenance of neuronal morphology. In addition, these protein structures affect important functions of the neuron, such as axon diameter regulation [57,58], conduction velocity [59,60], axonal transport [61,62] and synapse formation and function [63]. Therefore, it is not surprising that they have important relationships with pathological processes in addition to their physiological functions [64,65]. We decided to use NF as a marker to evaluate the development of new nerve fibers derived from mouse brain slices. In fact, NF and MAP2 both specifically stained nerve fibers and colocalized. Thus, we focused on NF only in the next steps.

NF forms the main material of the intermediate filaments in neurons. It has been shown that there are deteriorations in both the synthesis and function of this protein in AD [66,67,68], making it an important marker for revealing neuron damage. Although many mechanisms that cause neuron damage and loss in AD have been proposed, no hypothesis has been able to fully explain the etiopathogenesis to date. It is still controversial, how Aβ or tau causes the neuronal damage in AD [69]. It has long been well known that axonal dystrophy is observed in the early stages of AD [70] and regeneration is not observed following axonal degeneration. This situation may be due to inflammatory cytokines as well as increased regeneration inhibitory factors in AD. Identifying these possible factors may make an important contribution to the early diagnosis of the disease. There are many studies on the role of Aβ and tau in axonal degeneration [71,72], but new studies are also needed on alterations, such as the increase in inflammatory factors that prevent regeneration, the decrease in growth factors or the loss of their receptors. In our study, the number of newly formed axonal fibers per mm in the plasma-free collagen group was higher than in the groups containing healthy, MCI and AD plasma. These results suggest the existence of factors that suppress the formation of new nerve fibers in human plasma [73].

When the average lengths of newly formed nerve fibers extending from the lower edge of the slice along the lanes were compared between the groups, the average fiber length was higher in the group with plasma added from healthy individuals than in all other groups. This increase was statistically significant compared to the plasma-free and AD groups. Although there was a decreasing trend in the MCI group, the difference was not significant. When these results are evaluated, it can be considered that there are factors in the plasma that prevent the increase in the number of new fibers but promote the growth of existing ones. It can be suggested that factors in the plasma disabling the formation of new fibers increase (or the inhibitory factors decrease), and in addition, the growth of existing nerve fibers is suppressed. Identifying these factors and showing their relationship with the degree of the disease may provide very important parameters for early diagnosis of the disease.

### 4.3. MOG+ Dots on Nerve Fibers

Oligodendrocytes are glial cells located in the central nervous system and have important functions, such as producing myelin sheaths for axons, performing phagocytosis, and contributing to the integrity of the blood–brain barrier (BBB) [74]. Oligodendrocyte precursor cells (OPCs), which are the reserve population of these cells, can proliferate, differentiate and gain functionality in conditions such as physiological need or pathological damage. Their basic function is to provide support and insulation to neurons through myelination [75]. In addition to increasing the conduction speed in axons and providing insulation, myelination is also essential for the survival of neurons, and this can be clearly seen in demyelinating diseases [76,77]. Oligodendrocytes are also affected by the pathology created in the central nervous system (CNS) environment by AD and may lose their functions or even their vitality [78]. OPCs, whose differentiation abilities are affected, may lose their ability to transform into mature oligodendrocytes and myelination [79]. Although OPC proliferation and new myelin production initially increase compensatorily in response to pre-existing myelin loss, overall myelin amounts decrease since the increase cannot meet the need in AD [80]. In the immunofluorescence labeling we performed with MOG, we observed that the MOG+ dot-shaped structures in the close vicinity of new nerve fibers increased significantly in the AD plasma group compared to the healthy control group. This increase may have developed as an initial compensatory response to the damaging effects of some factors found in AD plasma and our findings are consistent with the literature [80]. Identification of these factors may provide us with useful markers for the early diagnosis of AD.

Due to the short two-week incubation period, a compensatory increase in the AD plasma group is an expected result. If the incubation period had been longer, this ratio between the groups could have reversed due to the loss of function and number of OPCs. The fact that MOG+ structures could not completely wrap the nerve fibers and remained in the form of dots may have been due to the shortness of the 2-week incubation period and the in vitro environment not being fully similar to in vivo conditions, such as absence of blood circulation and endocrine factors coming from the circulation.

The number of MOG+ dots in the plasma-free group was statistical significantly lower compared to the healthy plasma group. Therefore, the plasma may contain some factors that are essential for the production of MOG and related myelin by oligodendrocytes, which is supported by others [81,82]. The absence of these factors in slices cultured on microcontact prints devoid of plasma may have caused the formation of insufficient MOG+ dots. All plasma-containing groups contained increasing amounts of MOG+ dots with the severity of the disease. The mean number of MOG+ dots in all of these groups was higher than the plasma-free group. Identifying and isolating these factors can also serve as very useful markers for early diagnosis of the disease.

### 4.4. Mass Spectrometry

Using differential mass spectrometry and proteomics, we identified three putative biomarkers for AD selected by the key words “oligodendrocyte” or “axon” or “myelin”: aldehyde-dehydrogenase 1A1, alpha-synuclein and protein S100-A4.

**Aldehyde dehydrogenase 1A1** (ALDH1A1 or retinaldehyde dehydrogenase 1) belongs to the aldehyde dehydrogenase family [83] and these enzymes are responsible for the detoxification of aldehydes that occur in the brain [84]. ALDH enzymes have a function in many physiological and pathological processes, such as the conversion of vitamin A to retinoic acid, folate and betaine synthesis, neurotransmitter synthesis, stem cell proliferation and differentiation [85]. As a result of these metabolic processes in the brain, several toxic lipid peroxidation products are formed [86] and may act in the progression of AD through oxidative cell damage and Aβ-mediated neurotoxicity [86,87]. In order to counteract this toxicity, ALDH1A1 may play a role in the detoxification of these toxic aldehydes [88,89]. In fact, dysregulation of ALDH enzyme function and metabolism leads to cytotoxicity, oxidative stress, energy deficits, apoptosis and cell death [90]. There is clear evidence that in the brain ALDH1A1 is produced by astrocytes but not oligodendrocytes [91], however ALDH1A1 seems to be essential for oligodendrocyte maturation [92,93] and the synthesis of (astroglial) ALDH1A1 could be important for remyelination [94,95]. Interestingly, a decrease in the number of newly formed mature oligodendrocytes has been observed in aged (AD) brains and a dysregulation of ALDH1A1 may play a role [92,96,97,98]. Our mass spectrometry results identify ALDH1AA as a new putative biomarker in plasma and it will be interesting to measure plasma ALDH1A levels in AD patients. So far, no single publication has measured ALDH1A in plasma of AD patients (*PubMed ALDH1A1 and plasma and Alzheimer, 23.09.2024*). Definitely, our data support the hypothesis that plasma-derived ALDH1A may cause the detoxification of aldehydes in brain slices and may protect new nerve fiber growth and MOG+ dot formation.

**Alpha-synuclein (α-SYN**) is a neuronal protein encoded by the α-synuclein gene (*SNCA*) and is highly associated with the function of synapses [99,100]. There is clear evidence that α-SYN is affected in neurodegenerative diseases such as Parkinson’s disease, Lewy body dementia and multiple system atrophy (MSA) [101], but also familial and sporadic AD [102]. In the AD brain, tau and α-synuclein co-localize [103] and increased CSF α-SYN correlates with impaired cognitive function in MCI and AD [104]. This neurodegeneration is partly characterized by myelin loss and dysregulation of oligodendrocytes [105], and especially in MSA myelin loss and α-SYN deposits in oligodendrocytes are characteristic [106]. Increased α-SYN has neurotoxic effects, and mice with α-SYN overexpression display severe neurological damage and premature death [107]. Our data with mass spectrometry identify α-SYN as a putative plasma biomarker in AD and the results of our study are consistent with the literature. A PubMed search (*alpha-synuclein[title] and Alzheimer[title] and blood; 23.09.2024*) identified only three publications [100,101,102]. Our present study is in full line and may suggest that AD plasma-derived α-SYN protects nerve fiber growth and MOG+ dot formation in brain slices.

The **S100A4 protein** is a member of the Ca^2+^ binding S100 family and is encoded by the *S100A4* gene. It is particularly high in the CNS at neurogenesis and lesion sites and has been shown to cause neuroprotection and neuroregeneration [108,109,110]. S100A4 protein is also known to play a role in inflammation, and *S100A4* gene expression is increased in brain tissue in AD patients [111,112]. There are indications that S100A4 is a myelin-associated protein, especially because its expression is higher in myelinated nerve tissue [113,114] and it contributes to myelination [115]. However, there are not many studies on the direct relationship between S100A4 protein and oligodendrocytes, and mainly show an indirect interaction [116]. Our data with mass spectrometry identify the S100A4 protein as a putative plasma biomarker in AD. A PubMed search (*S100A4 [Title] and Alzheimer; 23.09.2024*) did not return a single publication. Thus, our present data are novel and may indicate for the first time that S100A4 protein could be a novel plasma biomarker for AD diagnosis. It will be interesting to generate recombinant S100A4 protein and microcontact print it and observe the direct effects on mouse-directed nerve fiber growth and MOG formation.

### 4.5. Limitations of This Study

There are some definite limitations in our study: (a) Microcontact printing of lyophilized plasma is a process that requires knowledge and experience and has a high probability of error. The results can sometimes be quite heterogeneous. To overcome this problem, we prepared multiple prints from each plasma sample and selected the best ones. We used a fluorescent control antibody (Alexa 546) to evaluate the print quality. (b) Since we used only a small portion of the plasma solution we prepared in collagen for printing purposes, we estimate that the amount of plasma protein on the membrane is at a low level of 10-100 ng/mL. These levels are far from the sensitivity of the ELISA method at picogram levels. (c) Due to their high potential for survival, 8–10-day-old mouse brain slices are widely preferred in the organotypic brain slice culture method. However, this preference reduces the model’s ability to represent a very complex disease seen in older ages, such as AD. The greatest obstacle to overcome this limitation is that a method with a high success rate for the culture of slices taken from adult animal brains has not yet been developed. (d) In our study, we used human plasma on mouse tissue. Due to the species difference, the reliability of our results is reduced to some extent. Nevertheless, we have to assume that the factors found in human plasma are effective due to their homological similarities between humans and mice. (e) The organotypic brain slice culture method we used has several difficulties. Even in samples within the same group, deviations in results can be seen from time to time. Although we have taken utmost care to ensure that the results are within certain standards in all slices, the development of evaluation methods with strict criteria to eliminate differences that may occur at the molecular level and not be reflected in morphology will increase the power of future studies. (f) The results of our study were interpreted by evaluating the protein expressions of neurons and oligodendrocytes. This method provides valuable information about the functions and pathologies of the relevant cells, but in our future studies, showing the changes in the gene expressions of these cells related to the investigated proteins will further increase the value of our hypothesis. (g) Although the 3D structure and cross-talk between cells are preserved most closely to the in vivo environment in the organotypic brain slice culture method, it is not possible to fully represent the in vivo environment due to axotomy, lack of blood circulation, and various disadvantages and deficiencies of in vitro conditions. (h) A two-week incubation period is insufficient to fully represent AD pathology, which is a chronic disease that lasts for years. Longer culture periods and monitoring of long-term changes will produce more information about the disease and diagnostic possibilities. (i) The donor plasma used in this study was collected over a period of many years. We have no information on the genetic profiles of the current donors that may be related to the disease (e.g., APOE ε4, PSEN1, PSEN2). Unfortunately, it is not possible to analyze and document the genetic backgrounds of these donors retrospectively. In future studies, the inclusion of these important data from donors must be considered. (j) Our study is technically a cross-sectional method. Indeed, a longitudinal method would provide more valuable data in such disease studies. However, since our study is a complex neurobiology experimental cell culture study, it was not possible to apply a longitudinal method. We believe that more advanced longitudinal studies should be planned in future experiments. (k) The application of invasive procedures and the use of long-term follow-up methods have not been included in our study due to ethical considerations. However, we hope that the reliable information to be obtained from similar studies will pave the way for invasive and long-term studies without ethical concerns. (l) Finally, it is not possible to fully transfer the study results to humans due to reasons such as being studied in a different species than humans, the animals studied being at a very young age unlike AD patients, and the extraordinary complexity of AD. However, the identification of molecules that can be used as early markers through special methods such as mass spectrometry can provide significant gains regarding the treatment and course of the disease.

## 5. Conclusions

In conclusion, we tested the hypothesis that combining the mouse organotypic brain slice cultures with the human plasma-loaded microcontact printing technique could serve as an early diagnostic tool for AD. After a two-week culture period, we observed new nerve fibers extending from the lower edges of the brain slices along the microcontact print lanes and the presence of MOG+ dots on their outer surfaces. While the formation of new nerve fibers decreased in AD plasma-loaded samples compared to healthy plasma samples, there was an increase in the number of MOG+ dots compared to the same group. We identified suitable novel biomarkers in the analysis performed by mass spectrometry method using the most characteristic samples of their groups. Using the optimal samples, we identified three putative biomarkers: aldehyde-dehydrogenase 1A1, alpha-synuclein and protein S100-A4. Thus, our strategy represents a novel and innovative approach to translate research findings from mouse models to human applications.

In light of the present study, we believe that in future studies, the development of ELISAs will be of importance to detect these three novel plasma biomarkers. Further, it will be important to generate recombinant proteins and test the effects in vitro on microcontact prints connected to brain slices. This should produce new information of pioneering value. We believe that this technique may have the potential to open new horizons that can be used in the diagnosis and treatment of several other neurodegenerative diseases of the brain.

## Figures and Tables

**Figure 1 biomolecules-14-01326-f001:**
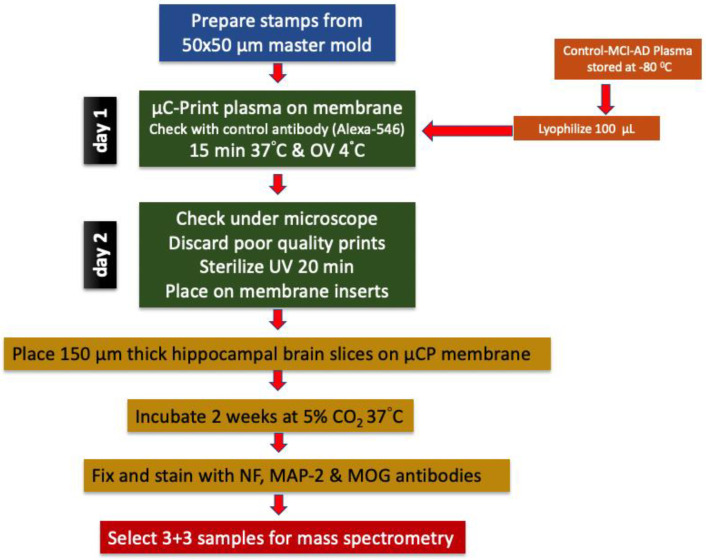
Experimental design of this study. Plasma samples from Alzheimer’s disease (AD) patients, mild cognitive impairment (MCI) patients and healthy controls (Co) were used for experiments. Mouse organotypic brain slices were coupled with plasma microcontact (µC) printed membranes and cultured for 2 weeks. At the end of the culture period, brain slices were fixed and labeled for neurofilament (NF), microtubule-associated protein-2 (MAP2) or myelin oligodendrocyte glycoprotein (MOG). For mass spectrometry analysis, selected plasma samples were used.

**Figure 2 biomolecules-14-01326-f002:**
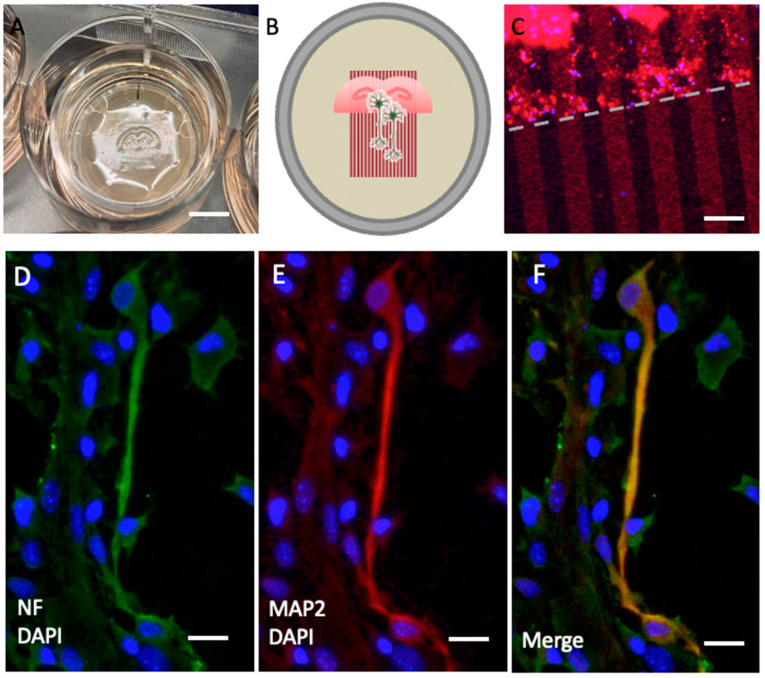
Immunolabeling of neurofilament (NF) and microtubule-associated protein-2 (MAP2). (**A**) Organotypic brain slices from the hippocampal level were cultured on an extra membrane in semipermeable inserts. (**B**) The scheme shows that these slices were connected to microcontact prints loaded with plasma, allowing nerve fibers to grow along them. (**C**) The microcontact prints can be visualized by additionally loading a red-fluorescent Alexa 546 antibody to assess the efficiency, appearing as red lines perpendicular to the border of the slice (dashed line). (**D**) Nerve fibers are stained with neurofilament (green, Alexa 488) and (**E**) with MAP2 (red, Alexa 546). (**F**) This panel shows the merged staining of NF and MAP2. The slices were counterstained with nuclear DAPI (blue). Scale bar = 1 cm (**A**), 75 µm (**C**), 25 µm (**D**–**F**).

**Figure 3 biomolecules-14-01326-f003:**
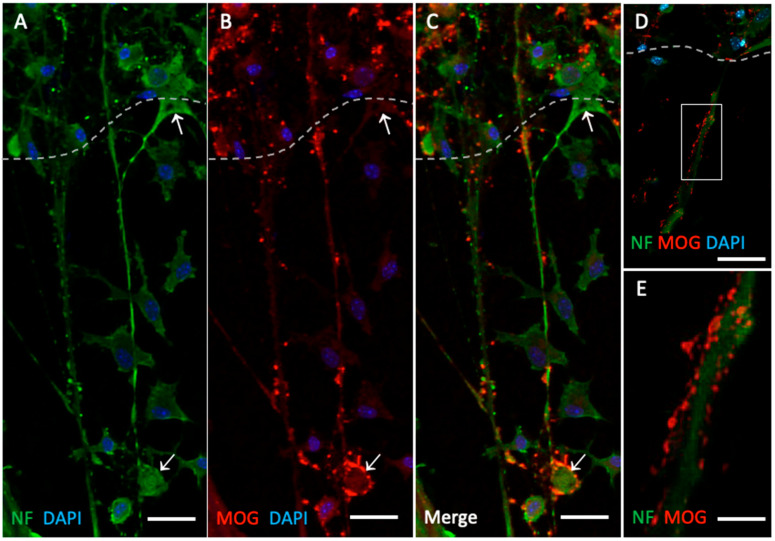
Immunolabeling of neurofilament (NF)-positive nerve fibers and myelin oligodendrocyte glycoprotein (MOG)-positive dots. Organotypic brain slices were cultured on plasma microcontact prints for 2 weeks. (**A**) Following fixation, they were stained for NF (green, Alexa 488) and (**B**) MOG (red, Alexa 546). (**C**) This panel shows the merged staining of NF and MOG. The slices were counterstained with nuclear DAPI (blue). The dashed white line indicates the border of the brain slices. The arrows point to specific examples of MOG-positive dots (bottom arrow) and negative staining (top arrow). (**D**,**E**) This panel shows an example at higher magnification of MOG+ dots located along NF+ nerve fibers. Scale bar = 75 µm (**A**–**C**), 100 µm (**D**), 20 µm (**E**).

**Figure 4 biomolecules-14-01326-f004:**
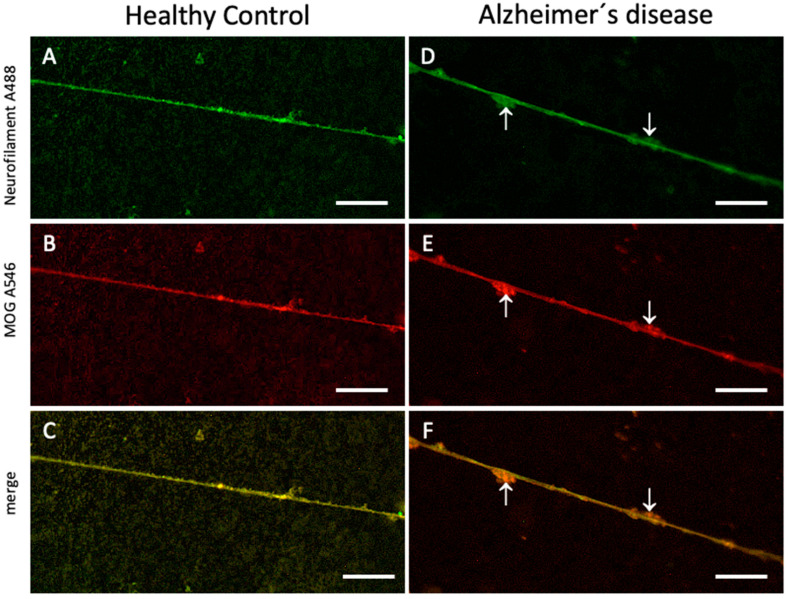
Immunolabeling of neurofilament (NF)-positive nerve fibers and myelin oligodendrocyte glycoprotein (MOG)-positive staining on healthy control plasma microcontact prints compared to those from Alzheimer’s samples. Organotypic brain slices were cultured on plasma microcontact prints for 2 weeks. (**A**,**D**) Following fixation, they were stained for NF (green, Alexa 488) and (**B**,**E**) MOG (red, Alexa 546). (**C**,**F**) These panels show the merged staining of NF and MOG. Note the MOG+ dots along the nerve fibers (indicated by arrows), which are more prominent and larger in the Alzheimer’s group. Scale bar = 20 µm (**A**–**F**).

**Figure 5 biomolecules-14-01326-f005:**
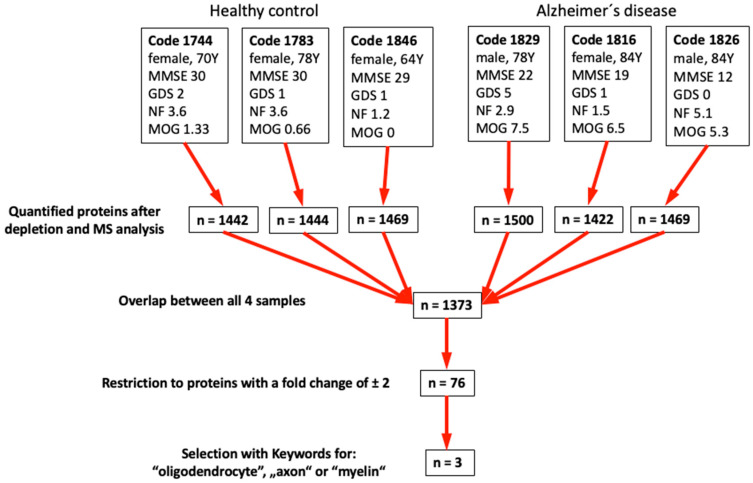
The best six plasma samples (three healthy controls and three Alzheimer’s disease) were selected based on the results of measurements and counting on NF+ axons and MOG+ dot structures. Proteins identified after depletion of 14 proteins are shown. The keywords “oligodendrocyte”, or “axon” or “myelin” were searched in combination with differential mass spectrometry results and ultimately three possible biomarkers were identified.

**Table 1 biomolecules-14-01326-t001:** Epidemiology of patients who participated in this study.

Patient	*n*	Male	Age	MMSE	GDS
Control	17	6	73 ± 1.5 vs.	29.6 ± 0.1 vs.	2.6 ± 0.7 vs.
MCI	18	6	75 ± 2.1	27.5 ± 0.1 *	3.7 ± 1.6
AD	17	6	82 ± 2.4 ***	18.3 ± 1.6 ***	2.5 ± 0.6

This table gives the patients used in this study, grouped to healthy controls and patients with mild cognitive impairment (MCI) and Alzheimer’s disease (AD). Values are given as mean ± SEM; *n* gives the number of patients; age is given in years. Statistical analysis was performed by one-way ANOVA with a subsequent Fisher LSD post hoc test compared versus (vs) the controls (* *p* < 0.05; *** *p* < 0.001). Abbreviations: MMSE, Mini Mental State Examination Scale; GDS, Geriatric Depression Scale.

**Table 2 biomolecules-14-01326-t002:** Quantification of neurofilament (NF)-positive nerve fibers and myelin oligodendrocyte glycoprotein (MOG)-positive dots.

Group	*n*	NF+ Fibers/mm	NF+ Fiber Length [µm]	MOG+ Dots per NF+ Fiber
µCP(-)	11	4.5 ± 1 **	217 ± 15.8 *	0.7 ± 0.3 *
Control	17	2.2 ± 0.3 vs.	383 ± 69 vs.	2.9 ± 0.4 vs.
MCI	18	2.6 ± 0.3	292 ± 30	4.2 ± 0.6
AD	17	3.3 ± 0.5	249 ± 21 *	5 ± 0.5 **

Brain slices were connected to microcontact prints of plasma-free collagen solution (µCP(-)) and from plasma of human healthy controls, or patients with mild cognitive impairment (MCI) or Alzheimer’s disease (AD). After 2 weeks in culture, slices were fixed and immunolabeled for NF and MOG. The number of NF+ fibers per mm, or the NF+ fiber length or the MOG+ dots per NF fiber are given. Values are mean ± SEM and *n* gives the number of samples. Statistical analysis was performed by one-way ANOVA with a subsequent Fisher LSD post hoc test and compared against the healthy plasma control (* *p* ≤ 0.05; ** *p* ≤ 0.01). As an additional control, slices incubated on plasma-free microcontact prints (PBS loaded to collagen solution) were evaluated.

**Table 3 biomolecules-14-01326-t003:** Putative biomarkers identified by differential mass spectrometry. The keywords “oligodendrocyte”, “axon” or “myelin” were used for the search.

Putative Biomarker Identified	Symbol	AccNr.	MW [kDa]	#Unique Peptides	Changein AD	Identity [%]	Similarity [%]
Aldehyde-dehydrogenase 1A1	ALDH1A1	P00352	54.8	13	2.1 ↑	87.0	94.4
Alpha-Synuclein	SNCA	P37840	14.5	8	3.6 ↑	95.0	97.1
Protein S100-A4	S100A4	P26447	11.7	5	2.5 ↑	93.1	98.0

This table gives the gene symbol, the accession number in the data bank (AccNr.), the molecular weight (MW), the number of unique peptides and the changes up (up arrow) or down (down arrow) in x-fold between healthy controls and Alzheimer’s disease patients. Homology levels between human and mouse were calculated via NCBI BLAST+ and are given as the percentage of characters that match exactly (Identity) and regions that may indicate functional, structural or evolutionary relationships (Similarity).

## Data Availability

The data that support the findings of this study are available upon request from the corresponding author.

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
