# Peer review of "From Organotypic Mouse Brain Slices to Human Alzheimer’s Plasma Biomarkers: A Focus on Nerve Fiber Outgrowth"

_biomolecules, 2024, doi:10.3390/biom14101326_

Round 1
Reviewer 1 Report
Comments and Suggestions for Authors
The purpose of this study was to find accurate biomarkers in blood plasma for AD early diagnosis. This is an interesting and important area of research. The paper is generally well written; however, the quality of the paper can be enhanced if the following points can be addressed.
1. How was the sample size determined? The study has a limited number of plasma samples, which can affect the generalizability of the results. A larger sample size would provide more robust data.
2. The authors did not mention if there is any diversity in the population of the study in terms of age, gender, ethnicity, or genetic background?
3. If the study is cross-sectional, it only provides a snapshot at one point in time. Longitudinal studies would be more effective in understanding the progression of AD.
4. There might be other factors influencing the results that were not controlled for, such as lifestyle, comorbidities, or medication use among participants. Can the authors briefly discuss or mention this if the data is available
5. Ethical considerations might limit the extent of the study, such as the ability to perform invasive procedures or long-term follow-ups but it could be ideal for proper conclusion.
6. The findings need to be replicated in independent studies to confirm their validity and reliability.
Addressing these limitations in future research can help improve the study’s robustness and the applicability of its findings.
Author Response
Response to the reviewers:
We would like to take this time to thank both reviewers for their fair critic of our manuscript. Below, we have provided a point-by-point response to their comments. Changes are highlighted in yellow.
Ad referee 1:
The purpose of this study was to find accurate biomarkers in blood plasma for AD early diagnosis. This is an interesting and important area of research. The paper is generally well written; however, the quality of the paper can be enhanced if the following points can be addressed.
- How was the sample size determined? The study has a limited number of plasma samples, which can affect the generalizability of the results. A larger sample size would provide more robust data.
Response: We fully agree that a larger sample size would provide more robust data. However, please imagine the complexity of the study. Plasma needs to be microcontact printed, connected to brain slices and cultured, then immunohistochemically stained and evaluated. A single set up contains a maximum of 3CO+3MCI+3AD which can be worked up within 4 weeks. The method we used to calculate the sample size of the study has been added in the text (Lines 272-274) and references (Reference 36).
- The authors did not mention if there is any diversity in the population of the study in terms of age, gender, ethnicity, or genetic background?
Response: We partly agree, as we already added age and gender in Table 1. The ethnicity was Caucasian and the genetic background (do you mean ApoE?) could not be measured. This is added to the Text (Line 118,119). The statement regarding the genetic background of donors has been added to the subsection of Limitations (Lines 647-652).
- If the study is cross-sectional, it only provides a snapshot at one point in time. Longitudinal studies would be more effective in understanding the progression of AD.
Response: Well, as already stated this is a neurobiology experimental cell culture study and not a simple clinical study; so a longitudinal study would be important but techically not possible, this has been added to Limits of the Study (Lines 652-657).
- There might be other factors influencing the results that were not controlled for, such as lifestyle, comorbidities, or medication use among participants. Can the authors briefly discuss or mention this if the data is available
Response: We agree, but you may know the complexity of such a complex human study to search for biomarkers. We have no information on lifestyle, comorbidities and detailed medication, expect statin use. This has been added to the Text (Lines 134-137).
- Ethical considerations might limit the extent of the study, such as the ability to perform invasive procedures or long-term follow-ups but it could be ideal for proper conclusion.
Response: Unfortunately, due to ethical concerns, it was not possible to use invasive procedures or long-term follow-up methods in our current study. This issue is stated under the Limitations (Lines 657-660).
- The findings need to be replicated in independent studies to confirm their validity and reliability.
Response: You are true, but a replication of the same set-up cannot be done by many labs worldwide, as we combine organotypic brain slices and microcontact printing of plasma, which is a single-standing tool. The next step is to develop and ELISA and to determine plasma levels, this can be replicated by others. This is discussed in Limits and Outlook (Lines 679-685).
Reviewer 2 Report
Comments and Suggestions for Authors
The manuscript titled “From Organotypic Mouse Brain Slices to Human Alzheimer’s Plasma Biomarkers: A Focus on Nerve Fiber Outgrowth” by Yilmaz, S.N.; et al. is a scientific work where the authors identified new biomarkers to identify the onset and progression of Alzheimer’s disease. This diagnostic tool could be expandable to other neurodegenerative disorders. This is a topic of growing importance and the manuscript is generally well-written. However, it exists some points that need to be addressed (please, see them below detailed point-by-point) to improve the scientific quality of the submitted manuscript paper before this article will be consider for its publication in Biomolecules.
1) The authors should consider to add the term “Immunofluorescence labeling” in the keyword list.
2) “Alzheimer’s disease (…) progressive loss of cognitive functions and memory (…) not possible to diagnose the disease early” (lines 33-37). Could the authors provide quantitative data insights according to the worldwide global burdens of this disease? This will significantly aid the potential readers to better understand the significance of this devoted research.
3) “However, when the diagnosis is made (…) by demonstrating post-mortem (…) neurofibrillary tangles in brain tissue” (lines 39-41). Here, even if I agree with this statement provided by the authors it may be desirable to mentioned how the redox conditions [1] and calcium concentration [2] directly impacts on the formation of these neurofibrils in different amyloidogenic proteins, respectively.
[1] Sasanián, N.; et al. Redox-Dependent Copper Ion Modulation of Amyloid-β (1-42) Aggregation in Vitro. Biomolecules 2020, 10, 924. https://doi.org/10.3390/biom10060924
[2] Carapeto, A.P.; et al. Morphological and Biophysical Study of S100A9 Protein Fibrils by Atomic Force Microscopy Imaging and Nanomechanical Analysis. Biomolecules 2024, 14, 1094. https://doi.org/10.3390/biom14091091
4) “Currently, four biomarkers (…) but it is not yet possible to study these biomarkers with a high percentage of accuracy in samples (…) and less risky” (lines 43-46). Here, some insights should be provided about the main reasons that lead the no possibility to monitor these biomarker in the aforementioned biological tissues.
5) “2.1. Human Plasma Samples” (lines 108-136). The sex of the patients who donated plasma should be remarked in this subsection to avoid any kind of sex-bias in this study.
6) Table 1 (line 284). The significant figures need to be homogenized. This comment should be taken into account in the rest of the main manuscript body text.
7) Figure 2, panels D-F (line 298). The lateral scale bar should be also indicated. Same comment for the Fig. 3, panels A-B and D-E (line 314), and the Fig. 4, panels A-C (line 360).
8) Discussion (lines 391-634).This subsection is clearly explained. No actions are requested from the authors.
9) Conclusions (lines 635-647). This section perfectly remarks the most relevant outcomes found by the authors in this work and also the promising future perspectives. It may be opportune to add a brief statement to discuss about the potential future action lines to pursue the topic covered in this research.
Comments on the Quality of English Language
The manuscript is generally well-written albeit it may be desirable if the authors could recheck it in order to polish those final details susceptible to be improved.
Author Response
Response to the reviewers:
We would like to take this time to thank both reviewers for their fair critic of our manuscript. Below, we have provided a point-by-point response to their comments. Changes are highlighted in yellow.
Ad referee 2:
The manuscript titled “From Organotypic Mouse Brain Slices to Human Alzheimer’s Plasma Biomarkers: A Focus on Nerve Fiber Outgrowth” by Yilmaz, S.N.; et al. is a scientific work where the authors identified new biomarkers to identify the onset and progression of Alzheimer’s disease. This diagnostic tool could be expandable to other neurodegenerative disorders. This is a topic of growing importance and the manuscript is generally well-written. However, it exists some points that need to be addressed (please, see them below detailed point-by-point) to improve the scientific quality of the submitted manuscript paper before this article will be consider for its publication in Biomolecules.
1) The authors should consider to add the term “Immunofluorescence labeling” in the keyword list.
Response: YES - we did so.
2) “Alzheimer’s disease (…) progressive loss of cognitive functions and memory (…) not possible to diagnose the disease early” (lines 33-37). Could the authors provide quantitative data insights according to the worldwide global burdens of this disease? This will significantly aid the potential readers to better understand the significance of this devoted research.
Response: We agree, thanks, this has been added to the Text. Over 55 million people worldwide are living with dementia, 60-70% are Alzheimer’s disease and the number is increasing. Relevant information and references have been added (Lines 37, 38) (References 1 and 2).
3) “However, when the diagnosis is made (…) by demonstrating post-mortem (…) neurofibrillary tangles in brain tissue” (lines 39-41). Here, even if I agree with this statement provided by the authors it may be desirable to mentioned how the redox conditions [1] and calcium concentration [2] directly impacts on the formation of these neurofibrils in different amyloidogenic proteins, respectively.
[1] Sasanián, N.; et al. Redox-Dependent Copper Ion Modulation of Amyloid-β (1-42) Aggregation in Vitro. Biomolecules 2020, 10, 924. https://doi.org/10.3390/biom10060924
[2] Carapeto, A.P.; et al. Morphological and Biophysical Study of S100A9 Protein Fibrils by Atomic Force Microscopy Imaging and Nanomechanical Analysis. Biomolecules 2024, 14, 1094. https://doi.org/10.3390/biom14091091
Response: Yes you are correct. We have added and discussed these two studies as references in the first sentence of section 1.3 (Lines 85-88) (References 21 and 22).
4) “Currently, four biomarkers (…) but it is not yet possible to study these biomarkers with a high percentage of accuracy in samples (…) and less risky” (lines 43-46). Here, some insights should be provided about the main reasons that lead the no possibility to monitor these biomarker in the aforementioned biological tissues.
Response: Yes we agree, we discuss this better. Obtaining CSF is invasive (lumbar puncture comes with health risks and costs) and there is no option for pre- or early disease screening tests. More detailed information on the subject has been added to the relevant section with references (Lines 47-52) (References 4, 5).
5) “2.1. Human Plasma Samples” (lines 108-136). The sex of the patients who donated plasma should be remarked in this subsection to avoid any kind of sex-bias in this study.
Response: We extended it, but it has already been added to Table 1. Gender information of the donors was added to the relevant sentence (Line 117-120).
6) Table 1 (line 284). The significant figures need to be homogenized. This comment should be taken into account in the rest of the main manuscript body text.
Response: Yes thanks, we homogenized all Tables and Figures.
7) Figure 2, panels D-F (line 298). The lateral scale bar should be also indicated. Same comment for the Fig. 3, panels A-B and D-E (line 314), and the Fig. 4, panels A-C (line 360).
Response: We agree, scale bars were added to all Figures panels.
8) Discussion (lines 391-634). This subsection is clearly explained. No actions are requested from the authors.
Response: We thank the referee for the critical revision and comments to improve the work.
9) Conclusions (lines 635-647). This section perfectly remarks the most relevant outcomes found by the authors in this work and also the promising future perspectives. It may be opportune to add a brief statement to discuss about the potential future action lines to pursue the topic covered in this research.
Response: We agree and add a brief statement on future actions on biomarkers: development of an ELISA and measure plasma levels, generate recombinant proteins and test the effects in vitro and/or load on microcontact prints.The relevant paragraph has been added to the conclusions subsection (Lines 679-685).
Round 2
Reviewer 1 Report
Comments and Suggestions for Authors
I do not have any further comments on the manuscript. The authors have addressed majority of the the concerns that I had and promptly made the revisions as per suggestion.
Reviewer 2 Report
Comments and Suggestions for Authors
The authors did a great deal of effort. For this reason, the scientific manuscript quality was greatly improved. Based on the significance and novelty of the gathered results I warmly endorse this work for further publication in Biomolecules.